# Recognizing Complex Activities by Combining Sequences of Basic Motions

Chenghong Lu , Wu-Chun Hsu and Lei Jing *

Graduate School of Computer Science and Engineering, University of Aizu, Tsuruga, Ikki-machi, Aizuwakamatsu 965-8580, Japan
* Correspondence: leijing@u-aizu.ac.jp

**Abstract:** For daily motion recognition, each researcher builds their own method to recognize their own specific target actions. However, for other types of target motions, they cannot use their method to recognize other kinds of motions because the features of their target motions that they extracted cannot be extracted from other kinds of motions. Therefore, we wanted to develop a general method that can be used in most kinds of motions. From our observations, we found that a meaningful motion is combined with some basic motions. Therefore, we could recognize basic motions and then combine them to recognize a target motion. First, we simply defined the basic motions according to the sensor's basic sensing directions. Second, we used k-nearest neighbors (KNN) and dynamic time warping (DTW) to recognize different categories of basic motions. Then, we gave each basic motion a specific number to represent it, and finally, used continuous dynamic programming (CDP) to recognize a target motion by the sequence of basic motions we collected. In our experiment on our basic motions, the accuracy of all of the basic motions is higher than 80%, so the recognition of basic motions is reliable. Then, we performed an experiment for recognizing the target motions. The results of recognizing the target motions were not good, the average accuracy being only 65.9%, and we still have to improve our system. However, we also compared our system with recognizing motions by using another general recognition method, KNN. And the average accuracy of using KNN to recognize motions was 53.4%. As this result shows, our method still obtains better results in recognizing different kinds of motions than using KNN.

**Keywords:** recognize motions; basic motions



## 1. Introduction

The recognition of daily motions holds increasing importance in the assessment of activities of daily living (ADLs), providing valuable insights into the quantity and quality of physical activity. Extensive research has been conducted in this domain, employing various methodologies and sensor technologies. For instance, Jeong et al. [1] utilized accelerometers for classifying daily postures and movements. Parate et al. [2] leveraged inertial sensors to identify smoking gestures, a method also applicable to analogous motions like eating. Totty et al. [3] combined muscle and inertial sensors for classifying ADLs, and Tang et al. [4] focused on hand motion recognition using muscle sensors.

Despite these advancements, a limitation persists in the specificity of each method to particular types of motions. For example, a technique effective in walking recognition may not be suitable for recognizing smoking or eating gestures, and vice versa. This segmentation in methodology underscores the need for a more universal approach to motion recognition.

Addressing these challenges, our research introduces a novel, comprehensive method for motion recognition. This method is grounded in the decomposition of complex activities into basic actions, enabling precise identification and categorization. Our approach simplifies the integration of new task actions by merely requiring the sequence of their

basic actions. Furthermore, it robustly accommodates variations in motion execution across different individuals, ensuring reliable recognition regardless of execution length or style.

Our research aims to develop a generalized method capable of recognizing a wide array of motions, encompassing everyday activities such as drinking water, smoking, opening doors, and eating with chopsticks. As illustrated in Figure 1, we observed that complex motions are typically composed of smaller, discrete movements. By deconstructing motions into their fundamental components and analyzing these smaller segments, we propose a novel approach for comprehensive activity recognition. To illustrate, consider the act of opening a door. If you want to open a door, you have to reach out your hand first, and then hold the doorknob, then turn the doorknob, and finally, open the door. Therefore, we could use the small pieces of the motions, such as reaching out our hand, moving our hand to the right or left, withdrawing our hands back, or spreading or holding some objects, making a fist, and so on. And then, combine these small motions to achieve activity recognition. For example, if we see the sequence of reaching out, holding some object, and turning the object, by this sequences of small motions, we can know that the person is opening a door. By using this basic idea, we could recognize most kinds of motions.

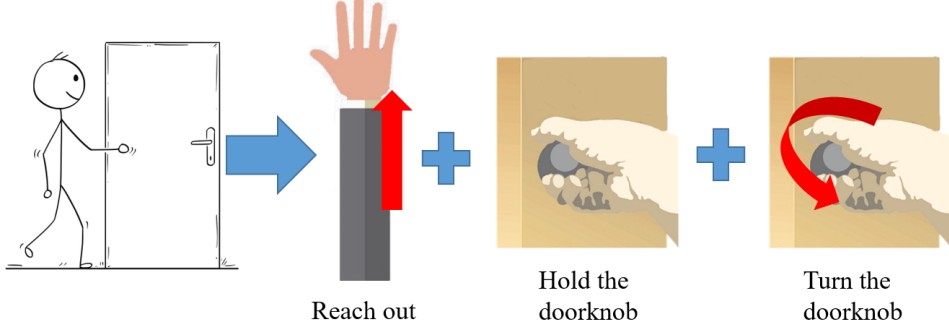

| Reach out | Hold the doorknob | Turn the doorknob |

**Figure 1.** Observation that the action of opening a door can be decomposed into a sequence of basic actions.

To facilitate this, we initially define a set of basic motions aligned with the axes of our sensors. For example, using a three-axis accelerometer, we can define six fundamental motions corresponding to the sensor's detection capabilities. The process then involves segmenting the most probable parts of these basic motions from continuous sensor data. We employ a thresholding technique for this segmentation, based on the premise that physical motions generate detectable energy or movement, manifesting as higher sensor values. Finally, to accurately classify these segmented basic motions, we utilize two distinct recognition methods: k-nearest neighbors (KNN) [5] for muscle sensor data, and dynamic time warping (DTW) for accelerometer and gyroscope data. This dual-method approach enables precise identification of the varied basic motions, laying the groundwork for our system's ability to recognize complex activities. Our research contributes a flexible, scalable, and accurate framework for recognizing a broad spectrum of daily activities. Then, when we obtain the basic motions, we have to combine the sequence of them. Here, we will first give each basic motion which we defined specific numbers to represent them. When we obtain the sequence of basic motions, we will transform the sequence of basic motions into a simple number list according to the numbers that we give. Then, we will use continuous dynamic programming (CDP) to recognize what kind of target motion is combined with these basic motions.

The contributions of this research are itemized below:

1. A two-stage recognition method was established that uses the inertial measurement unit (IMU) and electromyography (EMG) to identify basic actions and uses the sequence of basic actions to identify long actions.
2. The method of recognizing long actions by decomposing them into basic operations simplifies the addition of new actions and only requires the sequence of their basic

actions. It ensures recognition of actions despite variations in execution length across different individuals.

## 2. Related Work

In the past, there has been a lot of research on recognizing motions, which can be categorized into research on wearable sensors and ambient sensors. Various wearable sensors have been studied, such as inertial sensors [2,6], myoelectric sensors [7–11], bending sensors [12,13], and ambient sensors [14,15]. Using sensors for motion recognition generally involves steps including sensor data collection, preprocessing and feature extraction, and classification recognition. A comparison with related work is shown in Table 1.

**Table 1.** Comparison of related work.

| | Sensors | Research Target | Features | Algorithm |
|---|---|---|---|---|
| Our system | IMU, EMG | Daily activities | Basic motions | DTW, KNN, CDP |
| Ref. [2] | IMU | Smoking and similar motions | Find start and end features | RF, CRF |
| Ref. [3] | IMU, EMG | Category of daily motions | Statistics features (mean, SMA, RMS) | KNN |
| Ref. [4] | EMG | Hand motion | Multi-channel concordance correlation feature | Cascaded-structure classifier |
| Ref. [14] | Smart home | Daily activities | Based on which kinds of sensors | Dynamic sensor event segmentation algorithm |
| Ref. [16] | Acoustic sensor | Speech/text retrieval | CDP value | CDP, RIFCDP |

In terms of using the accelerometer to recognize motions, Ref. [1] used a three-axis accelerometer to classify the user's posture and movement to evaluate a person's activity quantity and ability. They used the accelerometer to recognize emergent situations such as falling during daily life. They used the differential signal vector magnitude (DSVM) to calculate the activity volume in daily life to recognize emergent situations such as falling and the state of activity. They could classify many kinds of daily motions. However, they only used an accelerometer to classify the user's posture and movement, they could not recognize more details about a person's gesture to detect more kinds of motions.

Ref. [2] used inertial sensors to recognize smoking gestures. They first used quaternion to calculate an arm trajectory and extracted features according to smoking gestures. Then, they used random forest (RF) to recognize smoking gestures and used a probabilistic model, condition random field (CRF), to analyze the sequence of smoking sessions. Their method could be used in other similar kinds of motions because the features they extracted are only for these kinds of motions, for example, they extracted the trajectory of the smoking gesture. However, for other kinds of motions, they have to re-collect or re-observe the motion data to extract some motion's representative features.

Daily action recognition is often used in health testing or rehabilitation. The research by Wan et al. [14] implemented extensive sensor networks in smart home environments. This method bases recognition on the location of sensor activation, with different activities inferred from the specific area of the sensor's installation. For instance, the activation of a sensor in the kitchen suggests activities like cooking or washing. While this approach effectively detects a variety of daily motions through sensors allocated to different household areas, it requires the extensive installation of sensors throughout the home. Additionally, this method is constrained to indoor environments, unable to track motions when users are outside.

In addition, for gesture recognition, many studies have used muscle sensors to detect user gestures. The study in [17] addresses the problem of noisy EMG signals and reviews the performance of various methods for analyzing EMG signals. Multisensory-integration-based EMG pattern recognition techniques have been developed. Although EMG signals

can provide extensive information about muscle movements, their amplitude is often influenced by various uncontrollable factors, such as electromyographic crosstalk, as mentioned in Ref. [18]. This can lead to misunderstandings about muscle activity, thereby affecting the accuracy of motion prediction. Ref. [19] addressed the segmentation issue of surface electromyography (sEMG) signals, proposing a standard protocol for segmenting and averaging muscle activations in cyclic movements.

Ref. [3] utilized a multimodal commercial sensor device incorporating sEMG sensors, an accelerometer, and a gyroscope to assess the feasibility of classifying categories of daily living activities as defined by the Functional Arm Activity Behavioral Observation System (FAABOS). KNN was employed to predict the FAABOS task categories. The inputs for KNN included the signal magnitude area (SMA) and mean values extracted from the acceleration and angular rate of change data, and root-mean-square (RMS) values computed from the sEMG data. This study utilized the same sensor device as ours. A straightforward experiment was conducted to determine which statistical features were more effective for use in KNN for classifying daily motions. However, their focus was solely on extracting statistical features to evaluate their system; they did not endeavor to recognize gestures or the forearm's trajectory.

Ref. [4] used a multi-channel surface electromyography sensor to recognize some hand motions. They extracted the multi-channel concordance correlation features which investigate the agreement between two signals. Then, they used a cascaded-structure classifier to recognize hand motions. The first step of the cascaded-structure classifier used energy ratio features to classify several separable groups. As the result of the first step, the similar features of hand motions were classified together. To separate each group of hand motions clearly, in the next step they used the concordance correlation features to recognize hand motions. They used a cascaded-structure classifier to improve the results of the classification and the features they extracted were really meaningful for the gestures.

The study in [9] introduces a novel real-time hand gesture recognition method using sEMG to decode motor unit activities for various motor tasks. The method involves segmenting EMG signals into motion-related segments and applying a convolution kernel compensation algorithm for real-time global EMG decomposition. This technique was tested on high-density EMG data from eleven non-disabled participants performing twelve hand gestures. However, the reliance on high-density EMG data could also limit its practicality in everyday applications due to the need for specialized equipment and setup. Ref. [20] demonstrated the feasibility of a surgical instrument signaling (SIS) gesture recognition system using sEMG signals acquired from a Myo armband, aimed at developing processing routines for remote or robotic surgical applications, though it currently only recognizes shorter movements.

There are applications of myoelectric sensors for motion capture using deep learning algorithms for recognition [21–24]. The recognition rate for short actions is very high, while long actions require more data training and the computational cost becomes higher. In Ref. [25], inspired by the significant successes of deep learning in image processing, the researchers treated raw sEMG signals as images and constructed a convolutional neural network (ConvNet) model. This approach of converting sEMG signals into images offers a novel perspective and methodology for understanding and processing EMG signals. In Ref. [26], they adopted a strategy of fusing global and local features to enhance dynamic gesture recognition rates by extracting both types of features from sEMG signals to improve gesture classification performance. In further research, addressing scenarios with limited data or the need to adapt to evolving tasks, the study in [27] developed a method capable of inferring the required outputs with only a few training examples, significantly easing the training burden and enhancing the practicality of the approach.

In terms of segmentation and recognition, Ref. [16] used their method CDP to automatically segment and recognize speech or text words. This method is really helpful for motion recognition because segmentation is one of the difficult problems for motion

recognition. If we could use this method to automatically segment and recognize motions, it would help us solve the problem with motion segmentation and recognition.

In our system, electromyography sensors are utilized to recognize the user's gestures, accelerometers to detect the movement of the user's forearm, and gyroscopes to assess forearm rotation. These motions are referred to as 'basic motions', which are invariably performed during the execution of a target motion. These basic motions are then employed as the defining features of our target motion. Finally, we combine them to achieve complete motion recognition. Due to the comprehensive nature of the basic motions' features, no information about the user's forearm movements or gestures is lost, enabling us to detect a wide variety of motions by amalgamating the recognition results from each sensor. Given that our system's motion features are based on basic motions, to recognize a new or undefined motion, one only needs to collect the basic motions associated with this new type. This approach obviates the need for time-consuming observation and collection of specific motion features.

### 3. Model

The organization of this paper is as follows. The research encompasses four parts, as illustrated in Figure 2. The first is the collection model, where we collect sensor data from the device, detailed in Section 4.3. Subsequently, these data undergo preprocessing on the computer. The second part is the segmentation model, elaborated on in Section 4.4, where preprocessed data are segmented to isolate basic motion components. In this model, accelerometer, gyroscope, and EMG data are used separately to segment various types of basic motions. The third part is the recognition model, the details of which will be explained in Sections 4.5 and 5. We aim to recognize the segmented parts obtained in the second model to identify the specific basic motions. In this model, two recognition methods are employed for identifying basic motions: KNN for hand status motions and DTW for both movement and rotation motions. The final part is the combination model, detailed in Section 6. Upon obtaining the sequence of basic motions from the above models, we combine them to recognize the target motion. In this model, each basic motion is assigned a specific number for representation, followed by inputting the sequence into CDP to recognize our target motions.

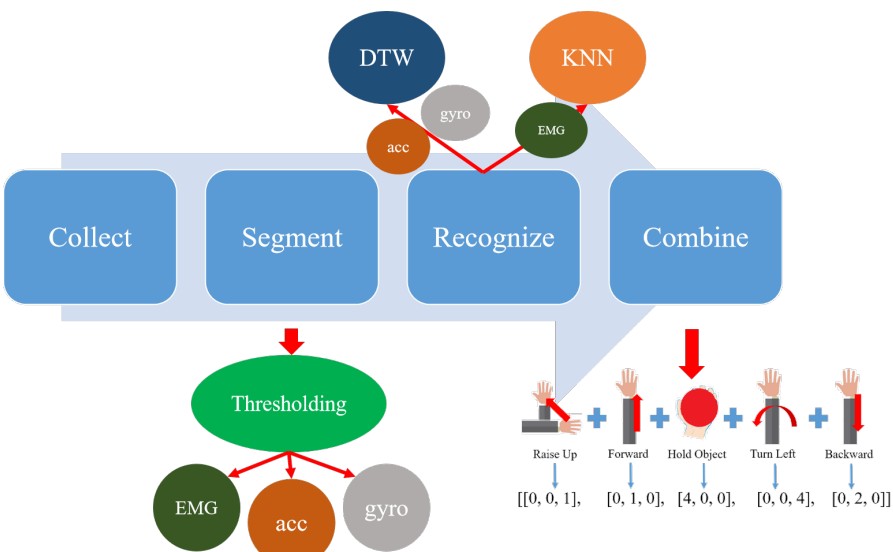

**Figure 2.** System flow.

### 4. System Architecture

The system architecture is depicted in Figure 3. Initially, input data are collected, followed by preprocessing of the raw data to facilitate easier segmentation. Subsequently, a predetermined threshold is applied to segment candidates of basic motion parts. Thereafter,

KNN is employed to classify the segments, thereby recognizing the hand status in basic motions. For recognizing the arm's movement and rotation as basic motions, DTW is utilized. In the final step, each basic motion in the sequence is assigned a specific numerical representation, and CDP is used to recognize the target motion.

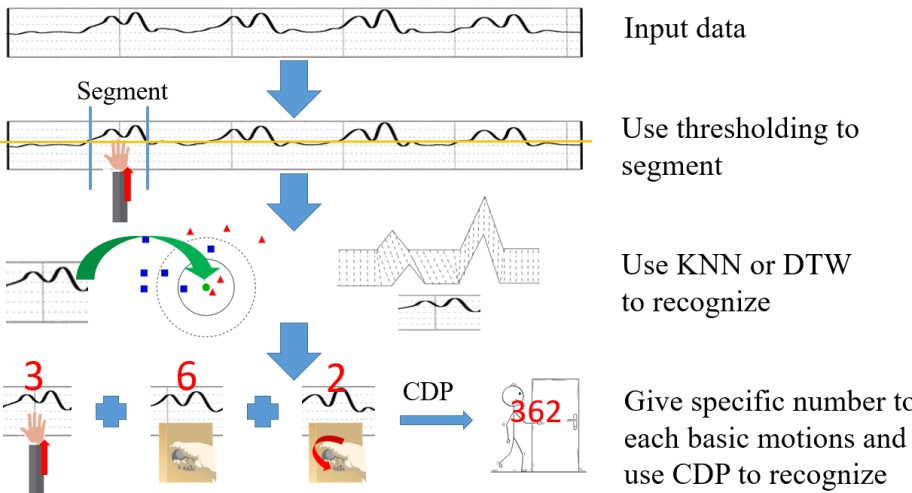

**Figure 3.** System architecture.

### 4.1. Device

For our sensor device, we utilize an armband called Myo, developed by the Canadian company Thalmic Labs in 2013. It is equipped with two types of sensors: muscle sensors and an IMU sensor. Myo features eight muscle sensors, as illustrated in Figure 4. The muscle sensors are capable of detecting whether the user's hand is exerting force, thereby enabling the detection of the user's gestures. The IMU sensor provides three types of inertial raw data: accelerometer, gyroscope, and quaternion data. This device will be used to detect the user's arm movement, rotation, and gestures using the different sensors.

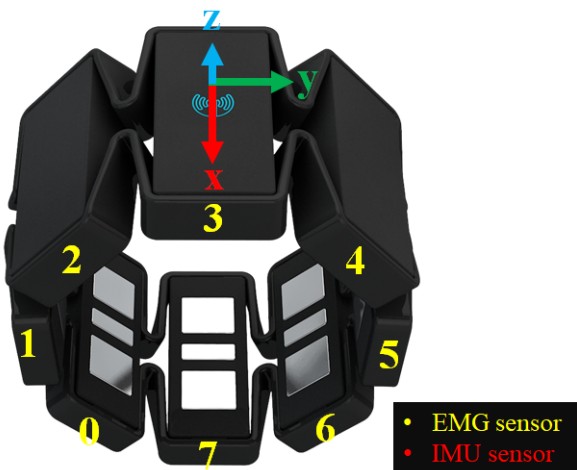

**Figure 4.** Sensor placement for Myo. (The yellow numbers represent 8 pairs of EMG sensors).

### 4.2. Definition of Basic Motions

In this section, we delineate the framework for basic motion classification within our study. Our methodology involves the categorization of motions into three distinct types, each aligned with a specific sensor modality: accelerometer, gyroscope, and muscle sensors. The accelerometer's output is instrumental in monitoring the translational movements of the user's forearm. Concurrently, the gyroscope data are pivotal for assessing the forearm's rotational dynamics. Finally, the integration of muscle sensor data is crucial for

the accurate interpretation of the user's gestural expressions. This tripartite approach forms the foundational basis for our motion detection and analysis system.

In the context of forearm movement, we have established definitions based on the directional output of the accelerometer, as illustrated in Figure 5. Specifically, the 'forward' and 'backward' basic motions correlate with the accelerometer's y-axis. Correspondingly, the 'move to left' and 'move to right' motions are associated with the x-axis of the sensor. Finally, the 'move up' and 'move down' motions are defined in relation to the z-axis of the accelerometer. This structured approach allows for precise motion categorization and enhanced interpretability of accelerometer data.

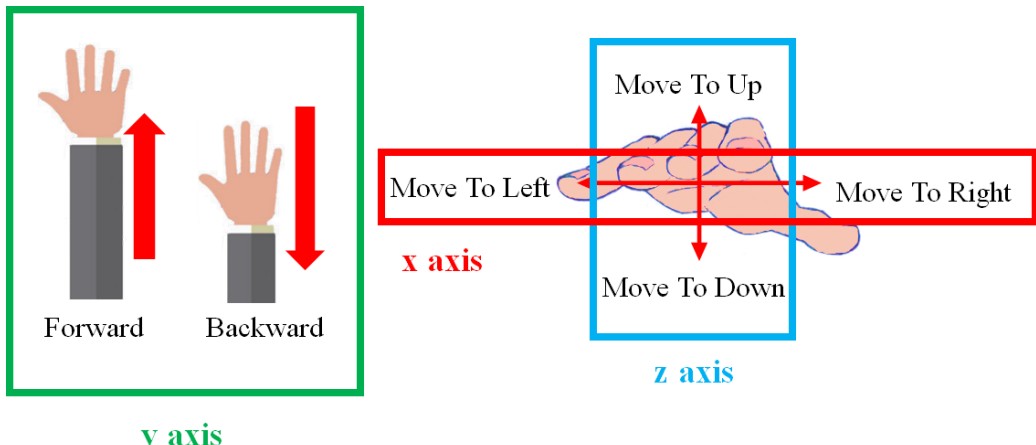

**Figure 5.** Movement of forearm.

For rotation of the forearm, we also define the basic motion according to the gyroscope sensing direction, as shown in Figure 6. "Raise Up" and "Put Down" correspond to gyroscope x sensing direction. "Turn to Left" and "Turn to Right" are for the y-axis. "Arm In" and "Arm Out" are for the z-axis.

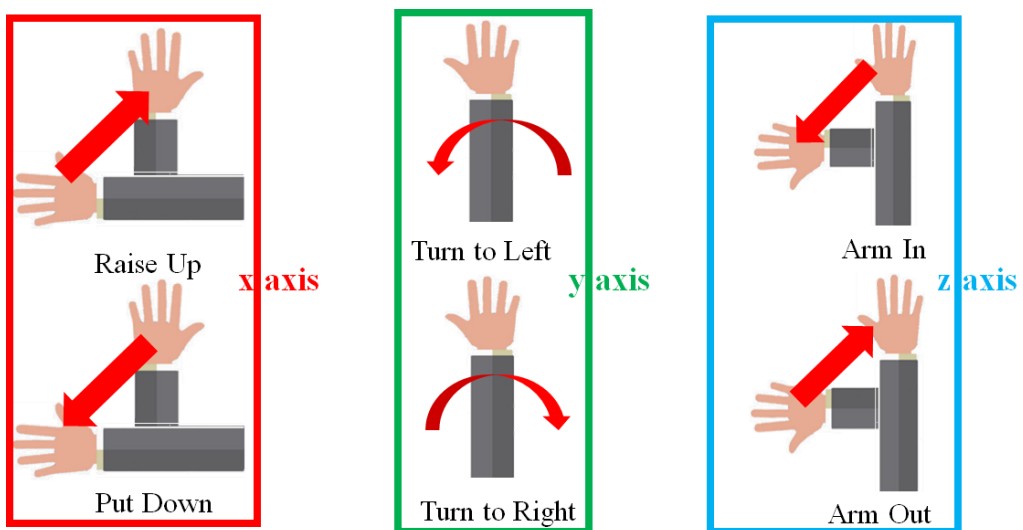

**Figure 6.** Rotation of forearm.

And for the status of the hand, we simply define five kinds of gestures, as Figure 7 shows: "hand out", "fist", "hand in", "open hand", and "hold object".

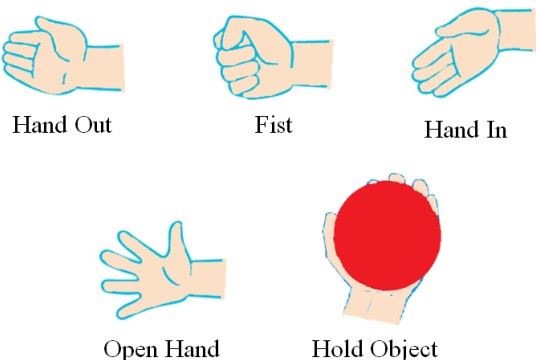

**Figure 7.** Status of hand.

*4.3. Data Collection and Preprocessing*

In the data collection phase, Bluetooth 4.0 was employed for communication with the Myo armband, utilizing a specific Bluetooth adapter compatible with Windows. The sensor data acquisition from Myo was facilitated through their Python API.

Once the sensor data were obtained, it was essential to preprocess these raw data to simplify the subsequent segmentation process. Given that thresholding would be applied to demarcate the basic motion segments, it was necessary to transform the data from each sensor into a one-dimensional format, thereby easing the segmentation of basic motions.

For the accelerometer and gyroscope data, a three-axis synthesis was performed using the formula outlined in Formula (1). This synthesis process effectively represents the user's movement and rotation; an increase in the synthesized values of the accelerometer or gyroscope data is indicative of the user initiating arm movement or rotation. Consequently, this provides a viable means to segment forearm movement and rotation through thresholding these synthesized values.

Regarding the EMG data, the root-mean-square (RMS) method, as depicted in Formula (2), was utilized to synthesize the sensor data. The RMS value of the EMG data reflects the extent of forearm exertion. As such, a higher RMS value corresponds to greater force exerted by the user's hand, thus enabling the effective segmentation of the exertion phase of the user's hand.

$$syn\_acc = \sqrt{x^2 + y^2 + z^2} \tag{1}$$

*syn_acc* refers to the three-axis synthesis of the accelerometer.

$$syn\_emg = \sqrt{\frac{\sum_{i=1}^{8} x_i^2}{8}} \tag{2}$$

*syn_emg* refers to the three-axis synthesis of EMG.

*4.4. Segmenting Basic Motions*

The primary objective of our research is to identify basic motions that form the foundation of most complex movements. This includes all forms of movement, rotation, and hand force exertion. As discussed in the preceding section, we hypothesize that threshold-based segmentation can effectively isolate each candidate of basic motions. Thresholding segmentation, a straightforward method, involves segmenting data based on a predefined threshold value. As illustrated in Figure 8, data exceeding this threshold is segmented and extracted. Accordingly, we apply this thresholding technique for the segmentation of basic motions. For the EMG data, we employ the RMS of the raw data to segment basic motions associated with the user's gestures, as previously mentioned. The optimal threshold for the RMS of the EMG data was determined through experimental trials, aimed at achieving effective gesture segmentation. In the case of IMU data, we utilize the root sum square of the raw data for segmenting basic motions related to the user's forearm. The thresholds for

the accelerometer and gyroscope data were similarly established through experimentation, with the goal of finding the most effective values for segmenting each type of sensor data.

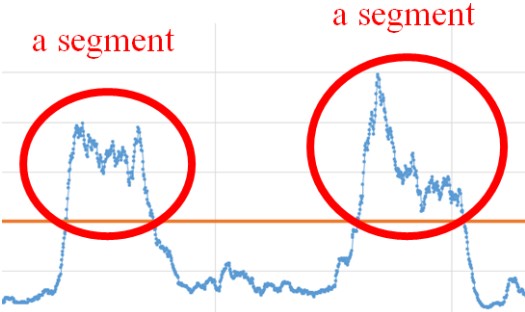

**Figure 8.** Thresholding segmentation.

*4.5. Recognition of Basic Motions of Gesture: K-Nearest Neighbors*

Following the segmentation phase, the focus shifts to the recognition of hand gestures. This process leverages the data from muscle sensors, which are each associated with specific muscle pairs. Different gestures activate varying combinations of these muscle pairs.

The KNN [28] algorithm serves as a relatively straightforward method for classification. The initial step involves training the KNN model by providing it with feature data and corresponding labels. During the testing phase, feature data are input into the KNN model, which then computes the distances between the input features and the training data. The classification result is determined based on the k-nearest neighbors in relation to the test data, effectively categorizing the gesture. The operational mechanism of the KNN algorithm is visually represented in Figure 9, providing an illustrative overview of its functionality in gesture recognition. Among the three closest points to the green dot, there are more red triangles, so we will classify them into the same category as the red triangles.

The KNN algorithm is characterized by its inherent simplicity and effectiveness, serving as a robust method for classification tasks. The process involves minimal complexity in training, which entails feeding the model with feature data and corresponding labels. Upon the introduction of new data, KNN demonstrates its adaptability, recalibrating itself to accommodate updated information. This feature makes it exceptionally suitable for dynamic applications where data are continuously evolving.

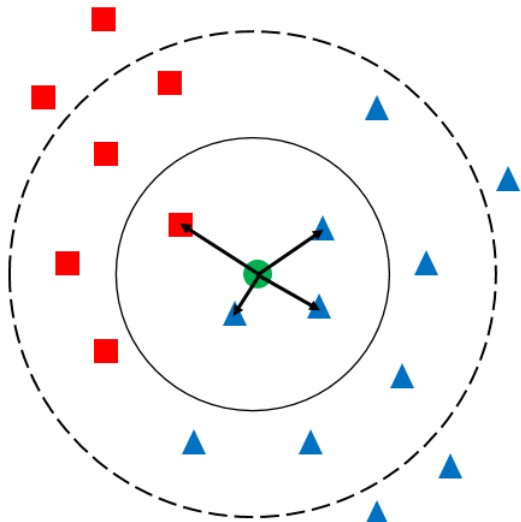

**Figure 9.** Operational mechanism of the KNN algorithm.

Consequently, we can input features from each sensor into the KNN model, as delineated in Formula (3). The rationale behind this approach is that the inputted features

effectively represent the activity of each muscle pair engaged by the user. In essence, the inputs to the KNN model are representations of the user's gestures. This expectation is predicated on the assumption that the correlation between specific muscle activations and corresponding gestures will be effectively captured and classified by the KNN model.

$$[RMS_{sensor_0}, RMS_{sensor_1}, RMS_{sensor_2}, RMS_{sensor_3},$$
$$RMS_{sensor_4}, RMS_{sensor_5}, RMS_{sensor_6}, RMS_{sensor_7}] \tag{3}$$

## 5. Recognition of Basic Motions of Forearm Movement and Rotation: Dynamic Time Warping

In the context of recognizing forearm movement and rotation, the directional properties of sensor values become particularly pertinent. The waveform patterns of sensor data are significantly representative of the arm's movement and rotation dynamics.

DTW [29] calculates the distance between reference data and test data. If the distance becomes smaller, it means the reference data waveform is more similar to the test data waveform. Figure 10 shows a schematic diagram of DTW.

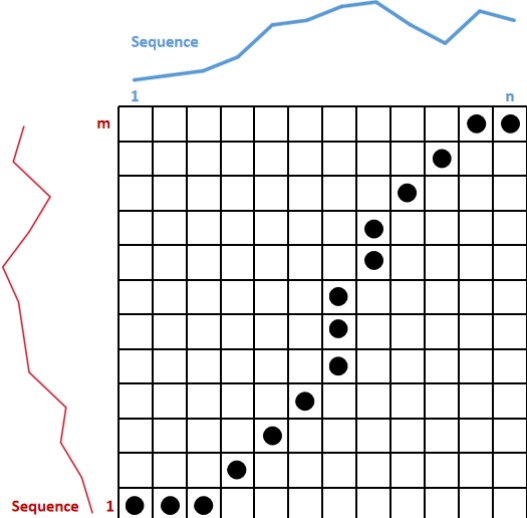

**Figure 10.** Schematic diagram of DTW calculating the distance between reference data and test data.

Hence, the application of DTW is proposed for identifying the basic motion that most closely aligns with the segmented part. This approach is predicated on the ability of DTW to discern patterns and similarities in time-series data.

DTW is distinguished by its remarkable ability to measure the similarity between two temporal sequences, even when these sequences vary in speed. This attribute of DTW lends it remarkable flexibility, vital for tasks requiring precise temporal alignment and sequence comparison. Furthermore, DTW's robustness against local distortions stands out, enabling it to efficiently process sequences that exhibit slight variations or contain noise. This resilience makes DTW a reliable tool for analyzing complex time-series data.

It is pertinent to note that both accelerometer and gyroscope data are inherently three-dimensional, encompassing the x, y, and z axes. Consequently, the input for the DTW algorithm comprises a sequence of three-dimensional data points. This multidimensional aspect of the data input is crucial for accurately capturing the complex dynamics of movement and rotation. The specific formulation and structure of this three-dimensional DTW input are detailed in Formula (4), which explains the methodology for transforming raw sensor data into a suitable format for effective motion analysis.

$$[[x_0, y_0, z_0], [x_1, y_1, z_1], \ldots, [x_n, y_n, z_n]] \tag{4}$$

## 6. Combining Basic Motions

After the identification of basic motions, the next critical step is their sequential combination. Given that the real-time sequence length of basic motions may vary from that of the training data, we employ DTW, previously introduced, to address these discrepancies in sequence length during the recognition of the target motion.

To effectively utilize DTW, a sequence of numerical values is required. However, at this stage, what we have are tags or labels representing basic motions. Thus, it becomes necessary to assign specific numerical representations to each basic motion. Considering the distinct nature of the three types of basic motions, we represent a basic motion as a three-dimensional data point. This representation includes the status of the hand, the movement of the forearm, and the rotation of the forearm. The structure of this representation is illustrated in Figure 11.

For example, as shown in Figure 12, the basic motion 'raise up', which pertains to a type of forearm rotation, is classified as the first type in the rotation category of basic motions. Therefore, it is represented numerically as [0, 0, 1]. This numerical coding system is consistently applied to all basic motions, following a standardized rule for representation, ensuring uniformity and precision in the encoding process.

The movements are divided into three categories: status of hand, movement of forearm, and rotation of forearm, and three dimensions are established for action label encoding.

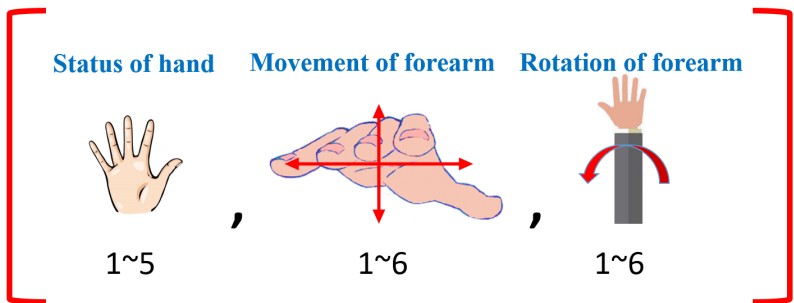

**Figure 11.** Action label encoding of basic motions.

Status of hand (1–5) contains five actions: fist, hand in, hand out, open hand, hold object.

Movement of forearm (1–6) contains six actions: backward, forward, move to up, move to down, move to left, move to right.

Rotation of forearm (1–6) contains six actions: raise up, put down, turn right, turn left, arm in, arm out.

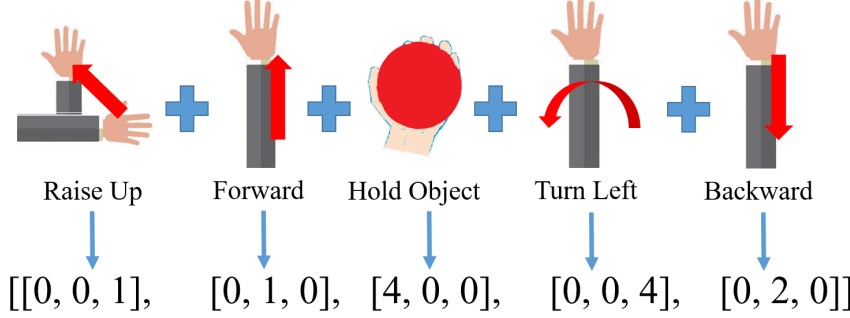

**Figure 12.** A door-opening action is decomposed into five basic actions according to Figure 11. Raise up label encoding is [0, 0, 1], forward label encoding is [0, 1, 0], hold object label encoding is [4, 0, 0], turn left label encoding is [0, 0, 4], backward label encoding is [0, 2, 0].

Upon obtaining the real-time sequence of basic motions, we proceed to transform this sequence into specific numerical values that correspond to the predefined basic motions. This transformation enables the input of both the sequence of basic motions and the training data into the DTW algorithm for recognition purposes.

However, an issue arises when utilizing DTW to amalgamate these basic motions into a target motion. We observed that the initial basic motion performed by an individual often deviates from the first basic motion in the training dataset. This discrepancy is due to the likelihood of executing an unrelated basic motion prior to commencing the target motion, as illustrated in Figure 13. Such a variance can lead to significant errors in the DTW output, given that DTW relies on accurately identifying the start and end points of the motion. This challenge necessitates a refined approach to ensure the effective utilization of DTW in our motion recognition system.

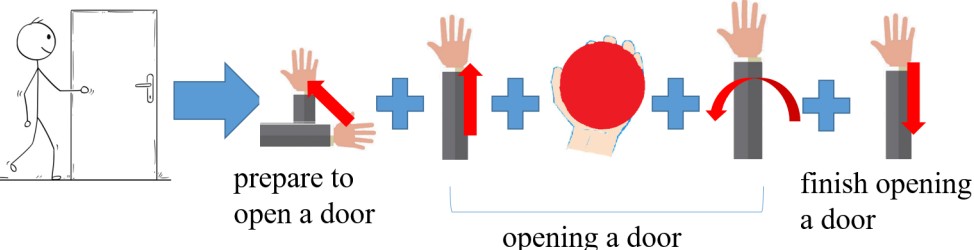

**Figure 13.** Observation of combining basic motions.

In response to the aforementioned challenge, we propose the application of CDP for the integration of basic motions. CDP, essentially a continuous variant of DTW, is adept at identifying the most similar segments within continuous data streams when compared to reference data. This capability makes CDP particularly suitable for our requirements.

The operational mechanism of CDP within our system is graphically represented in Figure 14. In this schematic diagram, the y-axis represents the input of reference data, while the x-axis corresponds to the test data input. By leveraging CDP, we anticipate overcoming the previously mentioned issue of misalignment between the initial basic motion in real-time data and the corresponding segment in the training data. CDP's ability to continuously analyze and match data segments holds significant potential for enhancing the accuracy and reliability of our motion recognition system.

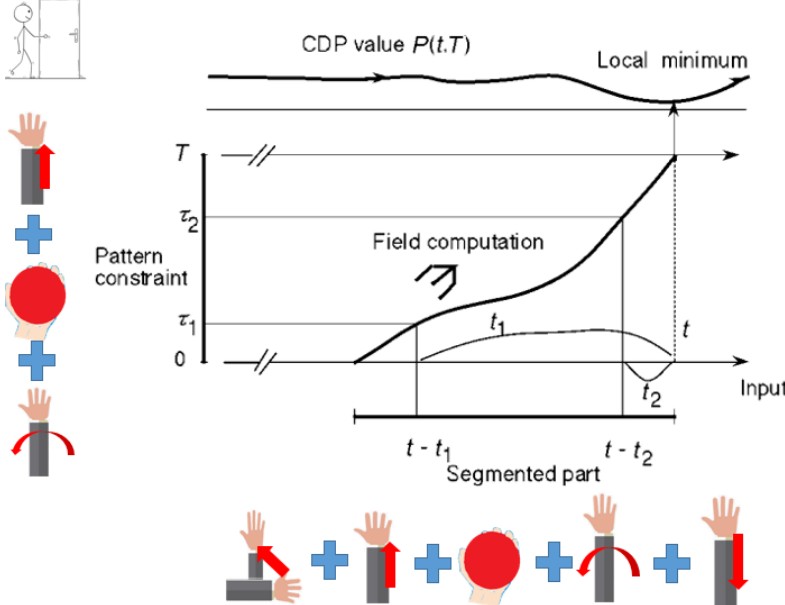

**Figure 14.** Example of CDP as used in our system.

## 7. Experiment

In this section, we will introduce our experiment environment and setting, then there is an experiment for basic motions which we defined, because before recognizing target

motions, we have to ensure that the accuracy of basic motions is high enough for them to be reliable features of target motions. In the end, we evaluate our system by comparing our method to recognize target motions with using KNN to recognize target motions.

### 7.1. Experiment Setting

For the experiment setting, we asked five subjects to help us collect basic motion training data. All the subjects were members of our laboratory. There are 17 kinds of basic motions. First, we demonstrated all the kinds of basic motions to each subject to let them understand how they should collect the training data. Next, we asked them to perform each kind of basic motion 10 times to save as text files to build our basic motions database.

After collecting all the subject's data, each kind of basic motion had 50 training data. If there were 50 training data for each basic motion, and for the movement and rotation of basic motions, our recognition method, DTW, needs too much time for computations; so, if there are too many training data, the system cannot perform the real basic motions result of the target motion, letting the target motion combine with some inaccurate basic motions. To solve these problems, we decided to use 20 training data for each basic motion to be our basic motions database. We then used 30 data for verification.

After constructing the basic motions database, we asked another three subjects in our laboratory to collect the training data of our target motions, drinking, opening a door, opening a drawer, and punching. The collection procedure was the same as the above procedure: we asked each subject to perform each target motion 10 times to build our target motions database. The special part of our target motions training data was the sequence of basic motions. For example, if we wanted to collect the training data of a target motion, like opening a door, we asked the subject to open a door to collect the training data, obtaining the sequence of basic motions like ['forward', 'open hand', 'hold object', 'turn right', 'backward']. And the sequence of basic motions would be our training data, not the real sensor data.

However, to compare our system with other methods, this time we saved the original sensor raw data as text files to input to other recognition methods to evaluate our system, and we also saved the sequence of basic motions that we acquired from the target motion as text files to build our system's target motions training data.

### 7.2. Evaluating Basic Motions

Now, we talk about the evaluation of the basic motions to ensure that the accuracy of basic motions is high enough to be the reliable features of target motions. We used leave-one-out cross-validation to evaluate our basic motions. For the basic motions of forearm movement and rotation, we used DTW to evaluate the result as mentioned in the previous section. And for the basic motions of hand status we used KNN to evaluate the result.

The experimental results pertaining to the basic motions of forearm movement are presented in Table 2. It is evident from the data that all movement-related basic motions achieved an accuracy exceeding 90%. Similarly, the results for basic motions of forearm rotation, as shown in Table 3, also demonstrate high accuracy for each categorized basic motion. Furthermore, the results for gesture-related basic motions, detailed in Table 4, indicate that the accuracy of these motions surpassed 80% in our experiments.

From these findings, it can be inferred that the recognition accuracy of basic motions is substantial. This high degree of accuracy validates the use of these basic motions as reliable features for the recognition of target motions in our system. The consistency of high accuracy across different types of basic motions underscores the robustness of our approach and its potential applicability in accurately identifying a diverse range of target motions.

**Table 2.** Basic motions of forearm movement.

| Predict / Real | Backward | Forward | Move_Down | Move_to_Left | Move_to_Right | Move_Up |
|---|---|---|---|---|---|---|
| Backward | 29 | 0 | 0 | 0 | 1 | 0 |
| Forward | 2 | 28 | 0 | 0 | 0 | 0 |
| Move_Up | 0 | 1 | 28 | 1 | 0 | 0 |
| Move_Down | 0 | 0 | 1 | 29 | 0 | 0 |
| Move_to_Left | 0 | 0 | 0 | 0 | 30 | 0 |
| Move_to_Right | 0 | 0 | 0 | 0 | 0 | 30 |
| Accuracy | 96.7% | 93.3% | 93.3% | 96.7% | 100% | 100% |

Average accuracy: 96.7%. Standard deviation: 3.

**Table 3.** Basic motions of forearm rotation.

| Predict / Real | Raise_Up | Put_Down | Turn_Right | Turn_Left | Arm_In | Arm_Out |
|---|---|---|---|---|---|---|
| Raise_Up | 30 | 0 | 0 | 0 | 0 | 0 |
| Put_Down | 0 | 30 | 0 | 0 | 0 | 0 |
| Turn_Right | 0 | 0 | 30 | 0 | 0 | 0 |
| Turn_Left | 0 | 0 | 0 | 30 | 0 | 0 |
| Arm_In | 1 | 0 | 0 | 0 | 29 | 0 |
| Arm_Out | 0 | 1 | 0 | 0 | 0 | 29 |
| Accuracy | 100% | 100% | 100% | 100% | 96.7% | 96.7% |

Average accuracy: 98.9. Standard deviation: 1.7.

**Table 4.** Basic motions of hand status.

| Predict / Real | Fist | Hand_In | Hand_Out | Open_Hand | Hold_Object |
|---|---|---|---|---|---|
| Fist | 28 | 1 | 0 | 0 | 1 |
| Hand_In | 0 | 30 | 0 | 0 | 0 |
| Hand_Out | 0 | 0 | 28 | 2 | 0 |
| Open_Hand | 0 | 0 | 4 | 24 | 2 |
| Hold_Object | 2 | 0 | 1 | 1 | 27 |
| Accuracy | 93.3% | 100% | 93.3% | 80% | 90% |

Average accuracy: 91.3%. Standard deviation: 7.3.

*7.3. Evaluating Target Motions*

In this section, we aim to assess the efficacy of our system through experiments involving four distinct target motions: drinking, opening a door, opening a drawer, and punching. For this evaluation, we employed the leave-one-out validation technique, ensuring that the test and training data consisted of sequences of basic motion tags. As outlined in the preceding section, these tags were transformed into specific numerical values to serve as inputs for CDP.

The target motions were then recognized using CDP. Additionally, to provide a comparative analysis of our system's performance, we also utilized the KNN method for recognition. KNN was selected as a comparative benchmark due to its ability to extract simple statistical features for recognizing a broad range of motions, unlike many specialized recognition methods. This comparison aims to highlight the versatility and effectiveness of our approach in contrast to more conventional methods.

In line with the findings from the study by [3], the optimal statistical features for KNN were identified as the mean values of the accelerometer and gyroscope data, along with the root mean square (RMS) of the EMG data. Consequently, we incorporated these three statistical features into the KNN model for our comparative evaluation. This approach is expected to provide a comprehensive assessment of our system's performance in recognizing diverse target motions, thereby validating the effectiveness of our methodology.

The results of applying our method to recognize target motions are presented in Table 5. The data reveal that the accuracy for the 'drinking' motion was 76.7%, 'open_door' achieved 46.7%, 'open_drawer' reached 53.3%, and 'punch' attained an accuracy of 86.7%. Notably, the results for 'open_door' and 'open_drawer' were less satisfactory. We attribute this to three primary factors.

**Table 5.** Experiment of recognizing motions by combining basic motions.

| Predict<br>Real | Drinking | Open_Door | Open_Drawer | Punch |
|---|---|---|---|---|
| Drinking | 23 | 0 | 2 | 5 |
| Open_Door | 4 | 14 | 6 | 6 |
| Open_Drawer | 0 | 1 | 16 | 13 |
| Punch | 2 | 2 | 0 | 26 |
| Accuracy | 76.7% | 46.7% | 53.3% | 86.7% |

Average accuracy: 65.9%. Standard deviation: 18.9.

Firstly, the specific numerical representations assigned to each basic motion may have contributed to the lower accuracy. In the context of CDP with sensor data, each value has inherent meaning and is indicative of the motion's sensitivity to the sensor. However, the numerical inputs in our method lack such meaningful relationships, potentially leading to significant errors in CDP's output.

Secondly, the issue may stem from the foundational definitions of basic motions in our system. While these motions are defined as simple, unidirectional movements, making them easily recognizable, this simplicity may not adequately capture the complexities of continuous, multidirectional target motions. As a result, our system might fail to accurately recognize certain aspects of these more complex movements.

Lastly, the limitations of our experimental device, which is worn only on the forearm, could have impacted the results. This setup precludes the detection of wrist rotation, a potentially crucial aspect of certain target motions. The absence of this information might have adversely affected the accuracy of our system in recognizing these specific motions.

The result of recognizing target motions using KNN is shown in Table 6. It can be seen that the accuracy of drinking is 80%, open_door is 40%, open_drawer is 36.7%, and punch is 56.7%.

**Table 6.** Experiment of recognizing motions using KNN.

| Predict<br>Real | Drinking | Open_Door | Open_Drawer | Punch |
|---|---|---|---|---|
| Drinking | 24 | 3 | 3 | 0 |
| Open_Door | 12 | 12 | 4 | 2 |
| Open_Drawer | 11 | 8 | 11 | 0 |
| Punch | 2 | 9 | 2 | 17 |
| Accuracy | 80% | 40% | 36.7% | 56.7% |

Average accuracy: 53.4%. Standard deviation: 19.8.

Comparing KNN with our system, as shown in Table 7, it is seen that our system obtains higher accuracy in most target motions, and only the accuracy of drinking is a little bit lower than the result of KNN. Though the results of our system should be better, by comparing it with KNN, we can see that our system is better than using some statistical features in KNN.

**Table 7.** Results of comparison of our system with KNN.

| Motions<br>Methods | Drinking | Open_Door | Open_Drawer | Punch | AVG (SD) |
|---|---|---|---|---|---|
| Our system | 76.7% | 46.7% | 53.3% | 86.7% | 65.9% (18.9) |
| KNN | 80% | 40% | 36.7% | 56.7% | 53.4% (19.8) |

## 8. Discussion

Our framework, designed for processing sequences of continuous actions, is marked by its flexibility and scalability. This feature makes it especially effective in addressing the complex challenges of movement recognition. Such an approach is highly beneficial in diverse areas including health monitoring systems, physical therapy and rehabilitation, and sports training and analysis. The framework's adaptability to evolving technologies further enhances its utility in these dynamic sectors.

Upon analyzing the factors contributing to the accuracy of our recognition results, we identified two primary challenges:

(i) Complexity in extracting fundamental actions: Extracting fundamental actions from complex, continuous motions is a significant challenge. In contrast to language, composed of discrete words and semantic blocks that inherently facilitate segmentation, physical movements are not naturally designed for conveying information. During action execution, there is no inherent focus on making these movements interpretable, which results in variability, even in identical movements. This inherent characteristic of actions adds complexity to the task of isolating fundamental actions from a continuous sequence.

(ii) Transitional movements leading to misclassification: The transitional movements that occur between basic actions, which are not included in the definitions of basic actions, frequently result in misclassifications. While these transitions contribute to the fluidity of movement, they do not conform to the established criteria of basic actions, thereby posing challenges in the recognition process.

To address these issues, our enhanced method undertakes a deeper analysis of the differences between basic actions within continuous actions and independent basic actions, aiming to improve the recognition rate of basic actions in continuous sequences. Moreover, transitional movements within these sequences are now recognized as a distinct category of basic actions, further refining our recognition process.

## 9. Conclusions

Our system employs a two-stage recognition model. Initially, it identifies basic actions, and subsequently, it utilizes the sequence of these basic actions to determine the target action. Basic movements have been categorized into three distinct types: forearm translational movement, forearm rotation, and hand state. To detect these movements, we use accelerometers for forearm translational movement, gyroscopes for forearm rotation, and muscle sensors for monitoring the hand state.

For the issue of how to segment the basic motions, we use KNN and DTW to recognize the basic motions in input data. For the issue of how to combine the sequence of basic motions, we combine the sequence of each basic motion by converting their specific number to the input of CDP and using CDP to recognize the target motion.

We propose to decompose a complex continuous action into a sequence of basic actions in three dimensions: status of hand, movement of forearm, and rotation of forearm. We build a two-stage framework for recognizing complex continuous actions. First, identify basic actions. Second, link basic actions to recognize complex continuous actions.

In our experiment, the accuracy of basic motions is good, but the accuracy of target motions is not very good. However, our system's accuracy of target motions is better than using KNN to recognize the target motions.

In the future, we will aim to further analyze the difference between basic actions in continuous actions and independent basic actions, and improve the recognition rate of basic actions in continuous actions. In addition, transitional actions in continuous actions are recognized as a type of basic action. And we should define more fine basic motions to let our system recognize more detailed motions. Then, we should install a sensor on the user's wrist to avoid losing the information on wrist rotation and movement to increase the accuracy of the target motions.

**Author Contributions:** Conceptualization, W.-C.H.; methodology, C.L. and W.-C.H.; software, W.-C.H. and C.L.; validation, C.L. and W.-C.H.; formal analysis, C.L.; investigation, C.L.; resources, L.J.; data curation, W.-C.H.; writing—original draft preparation, W.-C.H. and C.L.; writing—review and editing, C.L.; visualization, W.-C.H. and C.L.; supervision, L.J.; project administration, L.J.; funding acquisition, L.J., W.-C.H. and C.L. contributed equally to this paper. All authors have read and agreed to the published version of the manuscript.

**Funding:** This work was supported by JSPS KAKENHI, Grant Number 22K12114, the JKA Foundation, and NEDO Intensive Support for Young Promising Researchers, Number JPNP20004.

**Data Availability Statement:** The data presented in this study are available on request from the corresponding author.

**Conflicts of Interest:** The authors declare no conflicts of interest.

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
