# Peer review of "Recognizing Complex Activities by Combining Sequences of Basic Motions"

_electronics, doi:10.3390/electronics13020372_

Round 1

Reviewer 1 Report

Comments and Suggestions for Authors

The paper presents an innovative approach to motion recognition by decomposing complex activities into basic motions. It employs methods like K Nearest Neighbor (KNN) and Dynamic Time Warping (DTW) for motion categorization, along with Continuous Dynamic Programming (CDP) for recognizing target motions. The paper's strengths include its novel approach to motion recognition, clear methodological framework, and potential applicability in diverse fields. However, the paper could be improved in the following aspects:

1)Introduction and Literature Review: The introduction effectively sets the context for the research, but it could benefit from a more detailed discussion of previous work in this area. Including a broader range of literature would strengthen the argument for the necessity of the proposed method.

2)Methodology: The use of KNN and DTW is well-explained, but the paper could provide more justification for choosing these methods over others. Additionally, details on the dataset used, including its size and diversity, would help in assessing the robustness of the methodology.

3)Data Analysis and Results: While the results show promise, the accuracy rate of 65.9% indicates potential areas for improvement. The paper should discuss possible reasons for this level of accuracy and propose ways to enhance it.

4)Discussion: The discussion section is crucial for interpreting the results. The paper could expand on how the findings compare with existing theories and practices in the field. Discussing the implications of the research and potential applications would add depth to the paper.

5)Conclusion and Future Work: The conclusion succinctly summarizes the findings but could be expanded to include recommendations for future research based on the observed results and limitations.

Reviewer 2 Report

Comments and Suggestions for Authors

The article is interesting and worth further proceeding.

Before publication, I have a few suggestions to improve it.

The literature review is very poor. 10 references are definitely too little literature research, especially in relation to the last two years.

Figures 9 and 10 should be better described in the text with reference to the literature.

Fig. 11 and 12: what does e.g. 1-6 mean? in the Figure. This description is missing in the text. What is the difference between 1 and 2? What does 6 mean?

Fig. 12 - what is the difference between [0,0,1] and [0,0,4]? Where are these values used, How are they read by the system in the experiment?

There is too little detailed description of this methodology.

Fig. 14, part of the graphic obscures the description of the axis.

Symbols used in formulas should be described below them.

In conclusions, the sentences are written in the 1st person singular. Is the presented article the result of the work of one or three people as stated in the section about authors?

What are the main new developments presented in the work? This should be better articulated in the conclusions.

Reviewer 3 Report

Comments and Suggestions for Authors

The authors propose a recognize motions by combining sequence of basic motions. The authors conducted experiments on a real-world dataset and the experimental results demonstrate that the authors' proposed method outperforms.
However there are still plenty of problematic issues with this article, which will be itemized below.

-The authors effectively convey the motivation and problem, but the scientific background section lacks analysis and induction, appearing more as a compilation of information, for example don't use "I", use "we".
-The writing in the manuscript needs improvement as there are numerous mistakes that diminish its readability. The authors are advised to carefully review the manuscript and rectify all errors.
-If authors can include some works for comparison, it would be more impactful.
-Check the format of references which needs to be unified
-Discussion of results seems very limited here. Mathematical analysis is weak, needs more explanation.

-The authors conducted the relevant experiments on only one dataset and did not give statistical characteristics of the dataset

-what about  terms of evaluation metrics?

-The authors asserted that their method is possible tacked into account the temporal aspect of user behavior to make more accurate recommendations. To support this claim, the authors should incorporate a table detailing the time and memory usage for each step of their method. It is crucial to provide information on the duration of network training and the subsequent processing times.

Round 2

Reviewer 2 Report

Comments and Suggestions for Authors

Before publication I have still two minor suggestions

Fig. 12 - the description of the values used in the form [1,0,0] or [0,0,4] is still not clear to the reader. An explanation should be added in the figure caption.

The authors responded to comment 6, however, they did not complete the description in the text (Symbols used in formulas should be described below them.)

Reviewer 3 Report

Comments and Suggestions for Authors

The authors give all necessary answers

Author Response

Thanks for your time to review our paper.